# Generational Differences, Risk Tolerance, and Ownership of Financial Securities: Evidence from the United States

**Johnson Antwi** [1,*] **and Cephas B. Naanwaab** [2]

1    School of Financial Planning, Texas Tech University, Lubbock, TX 79409, USA
2    Department of Economics, North Carolina A&T State University, Greensboro, NC 27411, USA;
     cbnaanwa@ncat.edu
*    Correspondence: johnson.antwi@ttu.edu; Tel.: +1-806-742-5050

**Abstract:** This paper examines the relationship between generational differences, risk tolerance, and attitudes towards financial investments in a nationally representative sample from the United States of America. The sample consists of pooled cross-sectional data of three waves (2012–2018) and 80,000 observations from the National Financial Capability Study (NFCS). Using a probit model (with and without sample selection), all of the predictor variables are estimated to have statistically significant effects on the ownership of financial securities, with the expected sign effects. There is clearly a generational cohort effect, whereby Baby Boomers are on average more likely to own financial investments than Millennials, controlling for other factors such as incomes, education, and financial literacy. Generation Xers are statistically less likely to have investments in financial securities compared to Millennials. In general, Baby Boomers are more risk-averse and Generation Xers are more risk-loving than Millennials, accounting for education and income levels. The paper reveals a conundrum in which Baby Boomers (Gen Xers), although more (less) risk-averse, are more (less) likely to own financial securities. We control for reverse causality (endogeneity) in the relationship between risk tolerance and the ownership of securities, using the bivariate probit model. The level of financial knowledge of respondents correlates highly with asset ownership: individuals with high and medium levels of financial knowledge are more likely to own financial assets than those with low levels of financial knowledge. To address the limitations of the current findings with regard to generational attitudes towards financial investments, further research is recommended.

**Keywords:** financial securities; financial investment; generational differences; risk tolerance

**JEL Classification:** G11; G51

## 1. Introduction

The COVID-19 pandemic and the ensuing recession brought in its wake an increased sense of insecurity for many workers, as millions were forced out of the labor force, some probably into early retirement. Prior to the pandemic recession, most Millennials and Gen Xers planned to continue working even into their retirement. More than any other, these two generations undoubtedly bore the brunt of the Great Recession—high unemployment rates and low wages. This fact, coupled with high student debt burden, has created a feeling of less security among Millennials and Gen Xers, leading to a more conservative and cautious attitude when it comes to financial matters than with older generations (Kobler et al. 2015). Millennials are in general, less wealthy than earlier generations at the same age—even when controlling for educational levels and other factors (Kurz et al. 2018). In another study, around two-thirds of working Millennials were found to have nothing in their retirement savings (Brown 2018).

Recent studies on the aging American population have projected that Americans aged 65 and above will double from 49.2 million in 2016 to over 98 million by 2060 (U.S.

Department of Health and Human Services 2017). As different generational cohorts advance towards retirement, their investment decisions and their ownership of financial securities, or lack thereof, is gradually becoming a matter of concern. Malkiel (1996) suggested that the percentage of the portfolio allocated to stocks should decline as one approaches retirement. However, stocks offer a higher return on average than investments that are less volatile, such as Treasury Bills (Ibbotson Associates 1997). It is important to note that different generations might have different risk tolerances and varying perspectives regarding financial matters, and hence their ownership of financial securities might differ.

This research seeks to explore the relationship between generational differences, risk tolerance, and attitudes towards financial investments. Thus, the research question we seek to answer is, "Is there any difference in generational attitudes and risk tolerance with regard to the ownership of financial securities?". The study uses three waves of survey data from the National Financial Capability Study (NFCS 2012–2018). Roughly 27,000 individuals were polled in each wave from across all 50 U.S. states on a wide range of topics, from financial investments, financial risk tolerance, and financial knowledge, to demographic and socio-economic characteristics. Previous studies have observed that risk tolerance is inversely related with age. Even so, do all older individuals become equally risk-averse as they age? More broadly, what is the role of personal characteristics and the prevailing economic environment in this relationship? Other researchers also argue that the investment experience that is accumulated by older individuals over the years can make them risk-tolerant when it comes to investment, such that their portfolio values will not be negatively affected, even in old age.

The results that we present in this study indicate that Millennials are less likely to own financial investments than Baby Boomers, controlling for other important factors such as education, financial knowledge, and incomes. On the other hand, Millennials are significantly more likely than Generation Xers to own financial securities. Regarding the effect of risk tolerance on the ownership of investments, we find that risk-loving people are significantly more likely to invest in financial securities than risk-averse individuals, all things being equal. The results also reveal that highly educated, financially knowledgeable, and high-income individuals have a significantly higher likelihood of holding investments in financial assets. On average, our estimation also shows that Baby Boomers (Generation Xers) are more (less) risk-averse than Millennials. The paper reveals a conundrum in which Baby Boomers (Gen Xers), although more (less) risk-averse, are more (less) likely to own financial securities.

The remainder of the paper is organized as follows: Section 2 reviews the relevant literature pertaining to generational differences, risk tolerance, and attitudes towards financial investments. Section 3 presents the methodology, beginning first with the theoretical underpinnings of the model, and then lays out the empirical estimation strategy. Sections 4 and 5 present the data and empirical results, respectively. Finally, Section 6 provides concluding remarks and offers some limitations of the study, as well as suggestions for future research.

## 2. Literature Review

### 2.1. Generational Differences and Ownership of Financial Securities

The factors that influence financial risk tolerance and the ownership of financial securities have been examined by many different researchers using a host of different datasets and statistical techniques. However, there is scant literature on the generational differences in financial risk tolerance and the ownership of securities. Kim et al. (2019) investigated the role of financial knowledge in various short-term and long-term financial behaviors among Millennials in the United States. The analysis showed that financial knowledge is positively related with positive-performing short-term and long-term financial behaviors. Kadoya and Khan (2020) also examined the relationship between demographic and socio-economic factors and financial literacy in Japan by segregating financial literacy into financial knowledge, attitudes, and behavior, and providing a deeper understanding of these relationships.

The authors used data from the 2016 Financial Literacy Survey by the Central Council for the Financial Services Information of Japan and a linear regression model, and it was revealed that age is positively related to financial knowledge, but negatively related to financial attitude. This implies that middle-aged people in Japan are more financially knowledgeable, but that younger and older people are more positive regarding financial behavior and attitude.

Korniotis and Kumar (2011) analyzed the investment decisions of older individual investors, using a dataset of all trades and end-of-month portfolio positions of investors at a major U.S. discount brokerage house within the 1991 to 1996 time period. The authors used life cycle and learning models, and it was revealed that older and experienced investors are more likely to follow "rules of thumb" and experience that reflects excellent investment knowledge, but that older investors are less effective in applying their investment knowledge, especially if they earn a lower income, are less educated, and belong to a minority group. Similarly, Ansari (2019) showed that age is one of the most important factors that influence the decisions of investors. According to a study by Ameriks and Zeldes (2004), using pooled cross-sectional data from the Survey of Consumer Finances (SCF), and new panel data from the Teachers Insurance and Annuity Association-College Retirement Equities Fund (TIAA-CREF), the authors examined the empirical relationship between age and portfolio choice. Probit and Ordinary Least Square (OLS) models were adopted, and the results showed that there is no reduction in portfolio shares with age. Chovancová et al. (2019) compared various methods of country risk measurement, using the country risk of Italy as a case study. The analysis revealed that both the CPFER method and sovereign ratings show a similar level of country risk. Bailey et al. (2008) analyzed the individual investor decisions that motivate portfolio diversification into foreign stock markets. The authors showed that more experienced investors enjoy an informational advantage, and hence are more likely to invest internationally to have their portfolio diversified.

*2.2. Risk Tolerance*

Risk tolerance refers to the degree of risk that an investor is willing to bear in pursuit of a financial goal. There have been enormous contributions to the concept of risk tolerance in economics and finance literature. From the context of behavioral finance, risk tolerance can be thought of as being inversely related to risk aversion (Gilliam et al. 2010). Riley and Chow (1992) developed a model to examine the hypothesized relationships between risk tolerance and the given variables (individuals over age 65, those with incomes below the poverty level, and the very wealthy). The results revealed that relative risk aversion decreases significantly for the very wealthy, and also decreases with age up to a point, but increases after age 65 (retirement). Moreover, Wang and Hanna (2011) used 1983 to 1989 panel data from the SCF. The authors examined whether risk tolerance decreases with age. Using a heteroscedastic Tobit model, the results showed that age has a positive relationship with having an investment in risky assets when all other variables are held equal. Additionally, Spaeth and Peráček (2022) examined the proper use of cryptocurrencies, electronic securities, and security tokens in business. The authors applied the analysis of legal regulations and used case law and the analogy of law to critically point out selected application problems in a broader context. Yao et al. (2011) used the 1998–2007 Survey of Consumer Finances cross-sectional datasets and an analytical method to decompose the age effect on risk tolerance. The results revealed that risk tolerance generally decreases as people age. In addition, it was revealed that factors such as socioeconomic environments, perceptions, and demographic and economic characteristics all influence risk tolerance. Further, Geetha and Selvakumar (2016) concluded that the gender and income levels of investors are significantly influential in determining the risk tolerance of individual investors.

Grable (2000) conducted a study on financial risk tolerance and additional factors that affect risk-taking in everyday money matters. The author examined demographic, socioeconomic, and attitudinal characteristics that may be determinants of financial risk tolerance. The dataset consisted of 1075 faculty and staff members at a large southwestern university

in 1997. Respondents comprised 484 men and 591 women who were 20 to 75 years of age. In total, the average age of this sample was 43.50 years. Descriptive discriminant analysis and univariate test statistics were developed for this research. The results indicated that respondents who were older, male, more educated, and had more financial knowledge were more risk-tolerant. Rajnoha et al. (2016) showed that there is a relationship between business strategy and systems for measuring and managing corporate performance. These findings are important because it reflects positively on the achievements of the overall performance of companies. Hallahan et al. (2004) conducted an empirical investigation on personal financial risk tolerance. The authors used an Australian dataset with 20,415 respondents. Self-assessed risk tolerance (SRTS) and ProQuest survey and regression analyses were used, and the results showed that individuals underestimate their risk tolerance score. Additionally, age was found to be inversely related to risk tolerance.

In contrast with the above-cited studies, Palsson (1996) examined how the degree of relative risk aversion varies with household characteristics, and found that risk aversion increased with age among a cross-section of Swedish households. More so, a study by Morin and Suarez (1983) investigated households' demands for risky assets, using data from the asset holdings of Canadian individual households. Using analysis of covariance techniques, the authors discovered that on average, risk aversion increased with age, especially for those with a low level of net worth. On the contrary, risk aversion decreases with age for households with a high net worth. It was concluded that both age and net worth influenced risk aversion. Yao et al. (2004) studied the changes in financial risk tolerance from 1983 to 2001. The study used six SCF cross-sectional datasets from 1983 to 2001. Using a multivariate and cumulative logistic regression analysis, the results showed that an individual's risk tolerance fluctuates over time and is excessively influenced by recent events. Additionally, Fisher and Yao (2017) studied the gender differences in financial risk tolerance. Using the Survey of Consumer Finances data, it was revealed that individual variables such as income uncertainty and net worth relate to risk tolerance differently for men and women. In a similar study, Shusha (2017) examined the effect of demographic characteristics on financial risk tolerance among Egyptians. Using hierarchical regression analysis, the results demonstrated that there are significant effects from gender, age, educational level, and annual income on financial risk tolerance. A study by Nguyen et al. (2016) examined financial risk tolerance and its influence on investor decisions from within the context of financial advice. The results revealed that there is a positive relationship between client risk tolerance and investment decision-making.

All of the cited studies above have one thing in common: they analyze demographic and other economic factors that affect risk tolerance. These authors did not specifically address the impact of an individual's risk tolerance on investment decisions. This present study contributes to the literature with respect to how generational attitudes and risk tolerance affect people's choices to invest in financial securities.

## 3. Methods and Data

### 3.1. Random Utility Model

The theory of discrete choice (also known as the random utility model) posits that an individual makes a choice between the alternatives of 0 and 1 by picking the alternative that yields the highest level of satisfaction (utility). The discrete variable *y* takes a value of 1 if the individual deems the alternative 1 as yielding the higher utility, and 0 if the alternative 0 has the higher utility.

$$y = \begin{cases} 1, \; iff \; U_1 > U_0 \\ 0, \; iff \; U0 > U1 \end{cases}$$

Formulating the utility of these two alternatives in additive random utility form;

$$U_0 = V_0 + \varepsilon_0 \tag{1}$$

$$U_1 = V_1 + \varepsilon_1 \tag{2}$$

where $V_0$ and $V_1$ are the deterministic components of utility, and $\varepsilon_0$ and $\varepsilon_1$ are the random components of utility. If the utility of alternative 1 is greater than the utility of alternative 0 ($U_1 > U_0$), then the observed value of $y$ will be equal to 1 and the choice probability function is:

$$Pr(y = 1) = Pr(U_1 > U_0) = Pr(V_1 + \varepsilon_1 > V_0 + \varepsilon_0) = F(V_1 - V_0) \tag{3}$$

Assuming the utilities are a function of some explanatory variables, $x$, Equation (3) can be re-stated as;

$$Pr(y = 1) = F\left(x^{'}\beta\right) \tag{4}$$

where $F$ is the cumulative density function, $x$ is a vector of explanatory variables, and $\beta$ is a vector of unknown parameters. Assuming also that the density function follows the standard normal, then a probit model results as:

$$Pr = \Phi\left(x^{'}\beta\right) = \int_{-\infty}^{x^{'}\beta} \phi(v)dv \tag{5}$$

where $\Phi(.) =$ is the standard normal cdf.

The maximum likelihood estimator of the probit function has a log-likelihood function:

$$nL = \sum_{j \in S} w_j ln\Phi\left(X_j\beta\right) + \sum_{j \notin S} w_j ln\left\{1 - \Phi\left(X_j\beta\right)\right\} \tag{6}$$

### 3.2. Empirical Probit Model

For the purpose of developing the empirical probit model used in this paper, we assume that the decision-making unit (household or individual) has two choice alternatives: investment in financial securities (stocks, bonds, and mutual funds) or to not invest at all. From the foregoing discussion of the random utility model, if the decision maker's utility from investing (denoted as $U_1$) is greater than the utility from not investing (denoted $U_0$), then the probit function is stated as:

$$Pr(y = 1|X) = \Phi(X\beta) \tag{7}$$

where the dichotomous dependent variable, $y$, is ownership of financial securities ($y = 1$ if the household or individual has investments in securities, and 0 otherwise). The set of explanatory variables, $X$, included in the model are the education level of the decision maker, risk tolerance, financial literacy, home ownership, emergency fund, and age cohort.

### 3.3. Probit Model with Sample Selection

There are two possible sources of sample selection bias in this study. First, the respondents who chose to participate in the online survey might have certain characteristics in common, such as access to the Internet, and also the spare time to complete the survey. If this is the case, people who did not have access to the Internet may have been excluded from the survey; consequently, people with Internet access were self-selected to participate. The second source of selection bias is that individuals or households are likely to self-select into the ownership of financial securities through their participation in retirement programs. This may be due to their employment status and income levels. Those who are employed full-time are more likely to participate in retirement plans compared to part-time employees, and since most retirement accounts or 401(K) plans are overwhelmingly invested in financial assets, we suspect that there might be a sample selection problem.

To control for this selection issue, we estimate the probit model, accounting for sample selection (Van de Ven and Van Praag 1981). The probit sample selection model assumes that there is underlying latent function:

$$y_j^* = X_j\beta + u_{1j} \tag{8}$$

and that we only observe the binary dependent variable if the value of the latent function is positive.

$$y_j^{probit} = \left(y_j^* > 0\right) \tag{9}$$

Assuming that individuals self-select in the participation of the ownership of financial securities, then the value of the dependent variable is observable for persons that are selected, giving rise to the following selection function, where the selection depends on individual characteristics ($Z_j$):

$$y_j^{select} = \left(Z_j\gamma + u_{2j} > 0\right) \tag{10}$$

where $\mu_1 \sim N(0, 1)$; $\mu_2 \sim N(0, 1)$ $corr(\mu_1, \mu_2) = \rho$.

In the presence of selection bias, $p \neq 0$, applying the standard probit estimation, and therefore it yields biased estimates. The probit model with sample selection overcomes this bias, and yields consistent, asymptotically efficient estimates of all parameters of the model. We used participation in a retirement plan as the dependent variable of the selection equation: the reason being that participation in retirement plans is a form of self-selection into the ownership of financial securities.

### 3.4. Bivariate Probit Model

As an extension of the probit model, the bivariate probit estimates two jointly determined equations with correlated error terms, similar to a seemingly unrelated regression (Greene 2003).

The general specification of the bivariate probit model with two equations is given as:

$$y_1^* = x_1'\beta_1 + \mu_1 \tag{11}$$

where $y_1 = 1$ if $y_1^* > 0$ and $y_1 = 0$ otherwise,

$$y_2^* = x_2'\beta_2 + \mu_2 \tag{12}$$

where $y_2 = 1$ if $y_2^* > 0$ and $y_2 = 0$ otherwise,

$$Cov(\mu_1, \mu_2) \neq 0 \tag{13}$$

where $y_i^*$ is the unobservable or latent function that determines the outcome of the dependent variables $y_i$. If the two error terms are correlated, then independently estimating the equations separately produces biased estimates.

We jointly estimate two equations: the dependent variable for Equation (1) is the same as in the probit model described above ($y_1 = 1$ if the household has investments in securities, and 0 otherwise), and the dependent variable of the second equation is risk tolerance ($y_2 = 1$ if the individual/household has high financial risk tolerance, and 0 otherwise). The reasoning behind the use of a bivariate probit model is that risk tolerance and the ownership of financial securities are linked, and jointly estimating the two equations avoids bias.

## 4. Data

The dataset employed in this paper consists of pooled cross-sectional data of three waves (2012, 2015, and 2018) of the State-by-State National Financial Capability Study (NFCS). The state-by-state survey is conducted online every three years among a nationally representative sample of around 27,000 American adults. The main research objectives of

the NFCS are to benchmark the key indicators of financial capability, and to evaluate how these indicators differ with underlying demographic, behavioral, attitudinal, and financial literacy characteristics. The binary response variable that was used to estimate the probit models is whether the individual or household owns investments in stocks, bonds, and mutual funds (ownership of financial securities). The explanatory variables include the generational cohort (Millennials, Generation X, and Baby Boomers), the level of education, household income, willingness to take risks, home ownership, emergency funds, and levels of financial knowledge.

The below tables present the descriptive statistics based on the survey data. The mean age of the respondents is 47 years, with a standard deviation of 16.5 years. Generational cohort dummy variables (Millennials, Generation X, and Baby Boomers) are created based on the reported ages of the respondents. By definition, Baby Boomers are individuals who were born between 1946 and 1964, Gen X are individuals born between 1965 and 1980, and finally, Millennials are people born between 1981 and 1996. Since the dataset is not longitudinal, the construction of generational cohorts was conducted for each wave. The survey categorized the highest level of education that was completed by the respondent into seven levels: 1 (did not complete high school), 2 (regular high school diploma), 3 (GED or alternative credential), 4 (some college, no degree), 5 (associate's degree), 6 (bachelor's degree), and 7 (postgraduate degree). The mean level of education is 4.3, with a standard deviation of 1.67. This means that the average respondent has the educational level of an associate degree or a bachelor's degree.

The total household income data were collected into eight categories, ranging from 1 (less than $15,000) to 8 ($150,000 or more). Willingness to take risks (level of risk tolerance) was grouped on a Likert-type scale, into 10 levels, ranging from 1 (not at all willing to take risks) to 10 (very willing to take risks). The average risk tolerance level based on this scale is 4.95, with a standard deviation of 2.66. The level of financial knowledge was also assessed on a Likert scale ranging from 1 to 7, where 1 is very low financial knowledge and 7 is very high financial knowledge. The average financial knowledge of the respondents is 5.19 with a standard deviation of 1.28.

The survey respondents were also asked if they had set aside emergency or rainy-day funds that would cover expenses for three months, in the event of critical sickness, job loss, economic downturn, or other emergencies. This was a "yes or no" response, so the variable "emergency fund" was binary; 1 (has an emergency fund) and 0 (no emergency fund). Likewise, home ownership was assessed as a binary response: yes (the respondent or their spouse owned their home) and no (did not own their home).

Table 1b shows the descriptive statistics after the original variables from the survey were recoded to suit the purpose of this study. To make the dataset more meaningful to the current study, we constructed some of the variables from the original survey into different categories. We categorized age into generational cohorts (Millennials, Generation X, and Baby Boomers). Respondents in each survey wave who were born between 1981 and 1996 were classified as Millennials (28% of the total sample), those born between 1965 and 1980 as Generation X (35.3%), and those born between 1946 and 1964 were classified as Baby Boomers (36.7%). Additionally, the levels of education of the respondents were grouped into three (3): those who did not complete high school, those who graduated with a regular high school diploma, and respondents with GED or an alternative credential were classified as *high school education* (28.3%). Respondents with some college, no degree, an associate degree, or a bachelor's degree were classified as *college education* (62.6%), and finally those with post-baccalaureate education were categorized under *postgraduate education* (9.1%).

Respondents with an annual household income ranging from $0 to $34,000 were classified as having a *low household income* (33.9%), from $35,000 to $99,000 as having a *medium household income* (34.3%), and $100,000 and above were considered as having a *high household income* (31.8%). The degree of the financial risk tolerance of the respondent (willingness to take financial risks) was classified into three: *low-risk tolerance* (32.4%) if the respondent chose any number from 1 (not at all willing) to 3 on the original scale of 1–10.

Financial risk tolerance from 4 to 7 was considered as *medium-risk tolerance* (33.7%) and those from 8 to 10 (very willing) were classified as *high-risk tolerance* (30.8%).

**Table 1.** (**a**) Descriptive statistics based on the original survey data. (**b**) Descriptive statistics based on the variables constructed.

| (a) | | | | | | | | |
|---|---|---|---|---|---|---|---|---|
| **Variable** | **Obs** | **Mean** | **Std. Dev.** | **Min** | **First Quartile** | **Median** | **Third Quartile** | **Max** |
| Age | 80,164 | 47.1 | 16.56 | 18 | 33 | 48 | 61 | 101 |
| Level of education | 80,164 | 4.258 | 1.674 | 1 | 3 | 4 | 6 | 7 |
| Household income | 80,164 | 4.395 | 2.076 | 1 | 3 | 5 | 6 | 8 |
| Level of risk tolerance | 77,716 | 4.945 | 2.663 | 1 | 3 | 5 | 7 | 10 |
| Emergency fund | 76,789 | 0.482 | 0.5 | 0 | 0 | 0 | 1 | 1 |
| Retirement plan | 76,144 | 0.587 | 0.492 | 0 | 0 | 1 | 1 | 1 |
| Ownership of securities | 72,380 | 0.376 | 0.484 | 0 | 0 | 0 | 1 | 1 |
| Home ownership | 79,088 | 0.623 | 0.485 | 0 | 0 | 1 | 1 | 1 |
| Level of financial knowledge | 78,084 | 5.186 | 1.28 | 1 | 5 | 5 | 6 | 7 |
| (b) | | | | | | | | |
| **Variable** | **Obs** | **Mean** | **Std. Dev.** | **Min** | **First Quartile** | **Median** | **Third Quartile** | **Max** |
| Ownership of financial securities | 72,380 | 0.376 | 0.484 | 0 | 0 | 0 | 1 | 1 |
| Home ownership | 79,088 | 0.623 | 0.485 | 0 | 0 | 1 | 1 | 1 |
| Emergency fund | 76,789 | 0.482 | 0.5 | 0 | 0 | 0 | 1 | 1 |
| Millennials | 80,164 | 0.28 | 0.439 | 0 | 0 | 0 | 1 | 1 |
| Generation X | 80,164 | 0.353 | 0.442 | 0 | 0 | 0 | 1 | 1 |
| Baby Boomers | 80,164 | 0.367 | 0.482 | 0 | 0 | 0 | 1 | 1 |
| Low household income | 80,164 | 0.339 | 0.473 | 0 | 0 | 0 | 1 | 1 |
| Medium household income | 80,164 | 0.343 | 0.475 | 0 | 0 | 0 | 1 | 1 |
| High household income | 80,164 | 0.318 | 0.466 | 0 | 0 | 0 | 1 | 1 |
| Low financial knowledge | 78,084 | 0.087 | 0.282 | 0 | 0 | 0 | 0 | 1 |
| Medium financial knowledge | 78,084 | 0.489 | 0.5 | 0 | 0 | 0 | 1 | 1 |
| High financial knowledge | 78,084 | 0.423 | 0.494 | 0 | 0 | 0 | 1 | 1 |
| Low risk tolerance | 77,716 | 0.334 | 0.472 | 0 | 0 | 0 | 1 | 1 |
| Medium risk tolerance | 77,716 | 0.347 | 0.476 | 0 | 0 | 0 | 1 | 1 |
| High risk tolerance | 77,716 | 0.318 | 0.466 | 0 | 0 | 0 | 1 | 1 |
| High school education | 80,164 | 0.283 | 0.451 | 0 | 0 | 0 | 1 | 1 |
| College education | 80,164 | 0.626 | 0.484 | 0 | 0 | 1 | 1 | 1 |
| Postgraduate education | 80,164 | 0.091 | 0.288 | 0 | 0 | 0 | 0 | 1 |

In addition, respondents were asked about their level of financial knowledge. Those who chose financial knowledge on a scale from 1 to 3 were considered to have *low financial knowledge* (8.5%), respondents who chose from 4 to 5 were grouped as having *medium financial knowledge* (47.6%), and finally, those from 6 to 7 were classified as having *high financial knowledge* (41.2%). Investment in stocks, bonds, and mutual funds (*ownership of financial securities*), *emergency fund*, and *home ownership* were all constructed as dummy variables. For the ownership of financial securities, we assigned 1 if the respondent owned financial securities, and 0 otherwise. Out of the total respondents, 36.8% had some form of ownership of financial securities. Additionally, 62.3% of the respondents reported owning their home. Last but not least, 48.2% of the respondents reported having emergency fund savings.

## 5. Empirical Results

The goal of this study is to determine what factors drive individuals and households to make investments in financial assets (securities). In order to accomplish this goal, we developed and estimate a probit model (with and without sample selection) to explain the probability of the ownership of financial securities, conditional upon the respondent's demographic and socio-economic characteristics, such as age cohort (Millennials, Generation X, and Baby Boomers), level of education, risk tolerance, level of financial knowledge, home ownership, and having an emergency fund.

We assume that individuals or households make financial investment decisions by weighing the costs and benefits of owning financial assets. The costs of ownership of securities arise, in part, from the risks that are involved with either losing the value of or not earning any significant yields on these assets: There is a real chance of losing the

value of these assets, but there are also benefits in the form of asset appreciation and capital gains. For the average investor, if the perceived benefits of holding financial securities outweigh the costs, they will likely invest in them. Of interest then is: What factors drive some individuals to want to invest in financial assets, while others do not?

As already indicated in the foregoing section, the variables from the original survey were re-constructed to help to achieve the purpose of the present study. To that end, three dummy variables were constructed for age cohort, two of which (Generation X and Baby Boomers) were included in the models, with the third cohort (Millennials) left out as the reference group. For the level of education, two dummy variables were included for those with a postgraduate education and some college to bachelor's education, with the reference being those with a high school diploma or less. In a similar fashion, two dummy variables were included for household income: high household income and moderate household income (the reference being low household income); for risk tolerance, we included two dummy variables: high-risk tolerance and moderate risk tolerance (the reference group being low-risk tolerance individuals); for financial knowledge, we included two dummy variables: high financial knowledge and moderate financial knowledge (the reference being low financial knowledge).

Using the data presented in the previous section, we estimated the ordinary probit model described in Equation (7) and the sample selection probit model in Equation (10). Table 2 presents the results of the ordinary probit model, and Table 3 gives the results of the sample selection probit model. The difference between the two tables is that the probit model in Table 3 was estimated, accounting for sample selection, whilst Table 2 only presents the ordinary probit model, not accounting for sample selection. We anticipated that there would be a difference in the results presented in these two tables, and that any such differences could be attributed to sample selection bias. We do observe, based on the results, that there is only a small downward bias when estimating the ordinary probit model—the magnitudes of the coefficients and the marginal effects are slightly lower than those of the selection probit model.

**Table 2.** Probit model.

| | Probit | Probit |
|---|---|---|
| **Variables** | **Coefficients** | **Marginal Effects** |
| Home ownership | 0.400 *** | 0.1126 *** |
| | (0.0133) | (0.0036) |
| Emergency fund | 0.757 *** | 0.2130 *** |
| | (0.0116) | (0.0029) |
| Generation X | −0.205 *** | −0.0577 *** |
| | (0.0146) | (0.0041) |
| Baby Boomers | 0.0452 *** | 0.0127 *** |
| | (0.0130) | (0.0036) |
| Postgraduate education | 0.329 *** | 0.0926 *** |
| | (0.0209) | (0.0058) |
| College education | 0.213 *** | 0.0598 *** |
| | (0.0137) | (0.0038) |
| High household income | 0.646 *** | 0.1818 *** |
| | (0.0160) | (0.0048) |
| Medium household income | 0.310 *** | 0.0873 *** |
| | (0.0150) | (0.0042) |
| High risk tolerance | 0.678 *** | 0.1907 *** |
| | (0.0144) | (0.0039) |
| Medium risk tolerance | 0.399 *** | 0.1124 *** |
| | (0.0140) | (0.0039) |
| High financial knowledge | 0.328 *** | 0.0923 *** |
| | (0.0248) | (0.0048) |
| Medium financial knowledge | 0.109 *** | 0.0305 *** |
| | (0.0245) | (0.0068) |
| Constant | −2.091 *** | |
| | (0.0277) | |
| Observations | 69,903 | 69,903 |

Wald chi-squared statistic = 11602; *p*-value = 0.000; Pseudo R-sq = 0.25. Robust standard errors in parentheses.
*** *p* < 0.01.

**Table 3.** Probit model with sample selection.

| | Probit | Selection | Probit |
|---|---|---|---|
| Variables | Coefficients | Coefficients | Marginal Effects |
| Employed full-time | | 0.681 *** | |
| | | (0.0120) | |
| Home ownership | 0.413 *** | 0.447 *** | 0.1073 *** |
| | (0.0170) | (0.0122) | (0.0054) |
| Emergency fund | 0.753 *** | 0.328 *** | 0.1957 *** |
| | (0.0141) | (0.0119) | (0.0051) |
| Generation X | −0.156 *** | 0.101 *** | −0.04058 *** |
| | (0.0181) | (0.0142) | (0.0048) |
| Baby Boomers | 0.0636 *** | 0.209 *** | 0.01654 *** |
| | (0.0159) | (0.0128) | (0.0062) |
| Postgraduate education | 0.313 *** | 0.372 *** | 0.0814 *** |
| | (0.0245) | (0.0225) | (0.0084) |
| College education | 0.205 *** | 0.202 *** | 0.0533 *** |
| | (0.0172) | (0.0124) | (0.0054) |
| High household income | 0.760 *** | 1.100 *** | 0.1976 *** |
| | (0.0310) | (0.0159) | (0.0085) |
| Medium household income | 0.381 *** | 0.733 *** | 0.09907 *** |
| | (0.0273) | (0.0133) | (0.0081) |
| High risk tolerance | 0.694 *** | 0.0906 *** | 0.1804 *** |
| | (0.0173) | (0.0142) | (0.0065) |
| Medium risk tolerance | 0.398 *** | 0.116 *** | 0.1036 *** |
| | (0.0166) | (0.0131) | (0.0055) |
| High financial knowledge | 0.339 *** | 0.237 *** | 0.08806 *** |
| | (0.0314) | (0.0206) | (0.0099) |
| Medium financial knowledge | 0.103 *** | 0.209 *** | 0.0266 ** |
| | (0.0311) | (0.0196) | (0.0098) |
| Constant | −2.343 *** | −1.589 *** | |
| | (0.0550) | (0.0216) | |
| Observations | 71,768 | 71,768 | 71,768 |

Wald chi-squared statistic = 5989; *p*-value = 0.000; LR test of independent equations (rho = 0) = 49.33; *p*-value = 0.000. Robust standard errors in parentheses. *** *p* < 0.01, ** *p* < 0.05.

In both tables, we observe that nearly all of the explanatory variables are highly statistically significant. The models fit the data well with strong explanatory power, as indicated by the Wald chi-squared tests. Moreover, the sign effects of all coefficients are consistent in both models and they conform to expectations. The magnitudes of the coefficient estimates themselves, as is usually the case with any probit or logit models, are not very informative; thus, in what follows, we only focus on the interpretation of the marginal effects. The average marginal (or partial) effect of each explanatory variable is the change in the probability of the ownership of financial securities, P[y = 1], for a given change in the explanatory variable, holding the other explanatory variables constant. In the case of a dichotomous explanatory variable ($X_j$), the marginal effect is the change in the probability of the ownership of financial securities going from $X_j = 0$ to $X_j = 1$, while holding the other explanatory variables constant.

*5.1. Generational Differences and Asset Ownership*

The results of the ordinary probit (Table 2) and sample selection probit (Table 3) indicate that there are generational differences in the ownership of financial securities. Baby boomers (Gen Xers) on average are more (less) likely to hold investments in financial securities than Millennials. Holding all other factors constant, a typical Baby Boomer has a 0.045 (or 1.3%) higher odds of owning financial securities than the average Millennial (see Table 2). The Generation X cohort has lower odds, with approximately 5.8% less than the average Millennial in the ownership of financial assets. Once we control for sample selection bias, the Baby Boomers are 1.6% more likely than the Millennials to hold financial securities. The difference in the magnitudes between the ordinary probit and selection

probit is not much, but it is statistically significant and we attribute it to the sampling bias and other biases present in the data. Among the respondents in the survey, 37% were Baby Boomers, 35% Generation Xers, and 28% were Millennials. Thus, after accounting for the fact that Baby Boomers were slightly over-sampled, vis-à-vis Millennials, we still observe a clear positive tendency of the Baby Boomer generation to have financial investments in stocks, bonds and mutual funds than Millennials.

Previous empirical evidence varies regarding age and cohort effects on asset allocation. For instance, Yoo (1994), Poterba and Samwick (1997), and Heaton and Lucas (2000) found that investments in equities generally increase over most age brackets, and then gradually decline in retirement. By contrast, Ameriks and Zeldes (2004) suggested that there is no evidence supporting reductions in portfolio shares with age, nor any clear discernible cohort effects. What might account for the generational differences in attitudes towards financial investment? For sure, Baby Boomers are either in or close to retirement, and thus one would expect them to have more investments in financial assets, through 401(K) and IRA plans, than the typical Millennial. By logical extension, the same argument should apply to Gen Xers, who are older and should be more invested in financial assets than Millennials. Why then, are Gen Xers less likely than Millennials to hold financial assets? We argue that the difference has more to do with generational attitudes to money management and investments than just age differences. Baby Boomers, growing up in the difficult economic times of the 1940s and 1950s, may simply have acquired a completely different worldview of money and managing it, vis-à-vis the typical Millennial.

On the contrary, one could argue that most Millennials encountered the same harsh economic times as the Baby Boomers did, entering the workforce around the time of the Great Recession, with high unemployment rates and staggering student debts. By this reasoning, one would expect Millennials to have the same attitudes towards money management as the Baby Boomers. A study by the Center for American Progress found that Millennials are generally saving less than their parents did at the same age (Schwartz and Steinberg 2018). The study identified four major reasons for why Millennials are saving less: high youth unemployment, low middle-class wages, massive student debt, and poor-quality retirement plans. Indeed, most Millennials came of working age around the turn of the Great Recession and faced grim job prospects with attendant low wages. Even as of 2014—five years after the Great Recession ended—the unemployment rate among Millennials was still very high, at close to 15% (including those working part-time involuntarily), compared with an average unemployment rate of 6.5% in the general population. Sierminska and Doorley (2013) used a harmonized wealth data and a novel decomposition approach to show that cohort effects exist in the income profiles of asset and debt portfolios for a sample of European countries, the U.S. and Canada. The study revealed that younger households' asset participation decisions are more sensitive to income than older households.

What about Generation Xers? They were growing up in a different economic environment (more upbeat economic times) than either Baby Boomers or Millennials, and may not have the same attitudes to money management. It is puzzling as to why Gen Xers and Millennials appear to be comparable in terms of their ownership and investment in financial securities. To understand why generational differences alone may not tell the whole story of how people make investment choices, we turn now to other factors, starting with how differences in educational attainment could drive investments in financial assets.

*5.2. Educational Attainment and Asset Ownership*

On the question of how education affects the probability of holding financial assets, we compare two levels of educational attainment: those with a postgraduate education and college degree, and the reference group (a high school diploma). We find that highly educated individuals have a significantly higher rate of ownership of financial securities than less educated people. On average, people with postgraduate degrees are 9% more likely to own financial securities compared to the average person with only a high school

education or less. Those with college degrees are 6% more likely to own financial securities than non-college-educated individuals, a very significant effect, both in magnitude and in statistical significance. These magnitude effects are slightly lower (8% and 5%, respectively) in the sample selection model of Table 3. Grable (2000) concluded that individuals with a high level of educational attainment are more risk-tolerant than others, which explains our findings in this paper. Since highly educated individuals are more risk-tolerant, they are much more likely to own investments in financial securities, holding other things equal. Similarly, Kadoya and Khan (2020) found a positive association between education and financial knowledge. All else being equal, the higher the level of financial knowledge, the higher the probability of individuals having financial securities.

*5.3. Risk Tolerance and Asset Ownership*

It is probably safe to say that individual attitudes to financial risk is a major predictor of financial asset ownership. Holding other factors constant, risk-loving individuals (high-risk-tolerant) are 19% more likely to own financial assets than risk-averse (low-risk-tolerant) individuals. Once we control for sample selection bias, this difference is still robust, at 18%. The findings of this research tend to support the study by Xiao (1996), in which the author investigated the effects of income and life cycle variables on the ownership of 11 households that held financial assets. The results revealed that households in which the head indicated an average willingness to take risks were more likely to own most types of assets. In addition, Finke and Huston (2003) examined the relationship between net worth and financial assets and risk tolerance, using data from the 1998 Survey of Consumer Finances. The results revealed that a willingness to take financial risks is correlated significantly with a higher net worth. The authors further reported that the risk tolerance of those over age 65 is among the strongest predictors of a higher net worth.

*5.4. Financial Knowledge and Asset Ownership*

Another likely determinant of the ownership of financial assets is the degree of financial knowledge of the individual. All else remaining the same, people that are financially savvy are more likely to invest their money in stocks, bonds, and mutual funds. People whose self-reported financial knowledge is high are 9% more likely to invest in financial securities than those with low financial knowledge. For those with medium financial knowledge, they are 3% more likely than those with low financial knowledge to own stocks, bonds, and mutual funds. Additionally, Lusardi et al. (2017) showed that financial knowledge is a key determinant of wealth inequality in a stochastic life cycle model with endogenous financial knowledge accumulation. Further, they indicated that financial knowledge enables individuals to better allocate lifetime resources in a world of uncertainty and imperfect insurance, and in fact, 30–40 percent of retirement wealth inequality is accounted for by financial knowledge.

A study by van Rooij et al. (2011) concluded that financial literacy affects financial decision-making, in that individuals with a low financial literacy are less likely to have investments in stocks. Moreover, Brown and Graf (2012) used a novel representative survey to examine how financial literacy is related to household investment and borrowing among Swiss households. The authors reported that the respondents' financial literacy has a significantly positive association with their investment or financial market participation and their use of mortgage debt.

*5.5. Household Income and Asset Ownership*

With respect to family incomes, a positive correlation exists between incomes and the probability of investing in financial assets. Family incomes are by and large a stronger predictor of asset ownership than generational, educational, or financial literacy differences. High income households have an over 18% likelihood of owning stocks, bonds, and mutual funds, relative to low income households. A smaller effect (8.7%) is found for medium incomes relative to low incomes. This makes a lot of sense, since high net worth individuals

(or households) have more money to invest. Additionally, the result is in congruence with the research by Duasa and Yusof (2013), in which the authors showed that households with a high level of income tend to invest more in risky assets than households with a lower level of income. Moreover, Xiao (1996) indicated that income has positive effects on the ownership of financial assets.

### 5.6. Accounting for Endogeneity of Risk Tolerance

In the previous sections, we have analyzed the effects of risk tolerance on the ownership of financial securities, assuming that risk tolerance is exogenous. In this section, we allow for the possibility that risk tolerance is endogenous in this relationship, meaning that it is quite possible for a bi-directional relationship to exist between investment in financial securities and risk tolerance, making it necessary to estimate two equations jointly. Since these two dependent variables are binary, we account for endogeneity by estimating a bivariate probit model. The two binary dependent variables (ownership of financial securities and risk tolerance) are regressed on a set of exogenous variables. The error terms in the two equations are correlated, and therefore, we employ a seemingly unrelated regression estimation. The results of the bivariate probit model estimation are presented in Table 4. The average marginal effects are reported based on Equation (1) (ownership of financial securities). The estimated coefficients and marginal effects have expected signs and are statistically significant, with the exception of the Baby Boomers. By way of comparisons, the bivariate probit estimates are slightly different in magnitude from those that are obtained under the probit model (Table 2) and sample selection probit model (Table 3). The Generation X cohort has a slightly smaller effect, but this cohort still has a significantly lower likelihood of owning financial securities relative to Millennials. Baby Boomers (Generation Xers) have significantly less (more) risk tolerance than Millennials. Highly educated and high income individuals/households have a higher risk tolerance than less educated, low income individuals.

**Table 4.** Estimation of bivariate probit model.

| Variables | (1) Ownership of Financial Securities | (2) Risk Tolerance | (3) Marginal Effects of Equation (1) |
|---|---|---|---|
| Home ownership | 0.417 *** | −0.0143 | 0.0757 *** |
| | (0.0146) | (0.0116) | (0.0038) |
| Emergency fund | 0.841 *** | 0.307 *** | 0.1939 *** |
| | (0.0130) | (0.0106) | (0.0058) |
| Generation X | −0.201 *** | 0.0891 *** | −0.0263 *** |
| | (0.0159) | (0.0129) | (0.0040) |
| Baby Boomers | 0.00168 | −0.231 *** | −0.0279 *** |
| | (0.0190) | (0.0118) | (0.0047) |
| Postgraduate education | 0.397 *** | 0.264 *** | 0.1061 *** |
| | (0.0228) | (0.0198) | (0.0068) |
| College education | 0.252 *** | 0.152 *** | 0.06533 *** |
| | (0.0144) | (0.0115) | (0.0039) |
| High household income | 0.764 *** | 0.565 *** | 0.2112 *** |
| | (0.0276) | (0.0143) | (0.0073) |
| Medium household income | 0.362 *** | 0.211 *** | 0.0931 *** |
| | (0.0166) | (0.0125) | (0.0046) |
| Risk tolerance | 0.125 | | 0.0231 |
| | (0.154) | | (0.0241) |
| Constant | −1.731 *** | −0.253 *** | |
| | (0.0877) | (0.0132) | |
| Observations | 69,903 | 69,903 | 69,903 |

Wald chi-squared = 14154, *p*-value = 0.000; Wald test of rho = 0: chi-sq = 1.036, *p*-value = 0.308. Robust standard errors in parentheses. *** $p < 0.01$.

*5.7. Robustness Checks*

As a robustness check, we estimated the probit model using the original survey response data (see the results in Tables 5 and 6, below). The variables used here are based on the descriptive statistics in Table 1a. The reader will recall that the foregoing analysis was based on the constructed variables presented in Table 1b. The results presented in Tables 5 and 6 are in line with the expectation with regard to the significance and signs of the coefficients. In particular, these results show that age, household income, financial knowledge, and risk tolerance have positive and significant effects on the ownership of financial assets. These results are largely congruent with those obtained using the constructed variables in the preceding sections. In short, the findings are robust, whether we use the original variable definitions or those from the re-grouped variables. Another robustness measure is to compare and contrast the results using the self-reported financial knowledge of respondents with that of an objective measure. In this manner, we compared our estimate using the subjective measure of financial knowledge, defined as the individual's self-reported level of financial knowledge on a Likert scale, with another measure that is objective. The objective measured is constructed from the financial literacy test administered to the respondents to assess their objective financial literacy. The questions posed to respondents included things such as their understanding of interest rates, mortgage rates, inflation rates, differences between bonds and stocks, etc. Using principal component analysis, an objective index of the financial literacy score was constructed from the respondents' answers to these questions, and then used in the regression. Comparing the subjective financial knowledge coefficient and that of the objective measure shows a slight bias. In Table 5, the subjective financial knowledge coefficient is 0.057 (with a marginal effect of 0.0147), while in Table 6, the objective financial literacy coefficient is 0.0400 (with a marginal effect of 0.0122). Thus, using subjective financial knowledge, we find that there is a slightly overstated effect of financial knowledge/literacy on the likelihood of financial asset ownership. Individuals who self-report a higher degree of financial knowledge tend to have a higher tendency to invest in financial assets. This supports previous findings in the literature in which overconfident individuals are more risk-tolerant and therefore more likely to take financial risks.

**Table 5.** Probit model with original data (subjective financial knowledge variable included).

|  | (1) | (2) |
| --- | --- | --- |
| **Variables** | **Probit Coefficients** | **Marginal Effects** |
| Gender | 0.0518 *** | 0.0133 *** |
|  | (0.0115) | (0.0029) |
| Age | 0.0768 *** | 0.01967 *** |
|  | (0.00401) | (0.00102) |
| Ethnicity | −0.0868 *** | −0.0222 *** |
|  | (0.0136) | (0.00348) |
| Level of education | $-2.70 \times 10^{-5}$ | $-6.92 \times 10^{-6}$ |
|  | (0.00367) | (0.00094) |
| Household income | 0.127 *** | 0.0325 *** |
|  | (0.00356) | (0.0008) |
| Willingness to take risks | 0.0730 *** | 0.0187 *** |
|  | (0.00244) | (0.00061) |
| Emergency fund | 0.535 *** | 0.1369 *** |
|  | (0.0124) | (0.0031) |
| Home ownership | 0.227 *** | 0.0580 *** |
|  | (0.0145) | (0.0037) |
| Financial knowledge (subjective) | 0.0576 *** | 0.01475 *** |
|  | (0.00534) | (0.00137) |
| Marital status | 0.0203 *** | 0.0052 *** |
|  | (0.00522) | (0.00133) |
| Constant | −2.759 *** |  |
|  | (0.0399) |  |
| Observations | 73,318 | 73,318 |

Standard errors in parentheses. *** $p < 0.01$.

**Table 6.** Probit model with original data (objective financial knowledge variable included).

| Variables | (1)<br>Probit Coefficients | (2)<br>Marginal Effects |
|---|---|---|
| Age | 0.00808 *** | 0.00246 *** |
| | (0.000598) | (0.00018) |
| Gender | 0.0160 | 0.0048 |
| | (0.0168) | (0.0051) |
| Ethnicity | −0.00711 | −0.0022 |
| | (0.00819) | (0.0025) |
| Level of education | 0.0511 *** | 0.0156 *** |
| | (0.00559) | (0.0017) |
| Household income | 0.122 *** | 0.0372 *** |
| | (0.00504) | (0.0015) |
| Level of risk tolerance | 0.130 *** | 0.0396 *** |
| | (0.00356) | (0.00101) |
| Emergency fund | 0.813 *** | 0.2479 *** |
| | (0.0177) | (0.0048) |
| Home ownership | 0.322 *** | 0.0981 *** |
| | (0.0213) | (0.0064) |
| Financial literacy (objective) | 0.0400 *** | 0.01219 *** |
| | (0.00816) | (0.00248) |
| Constant | −2.883 *** | |
| | (0.0623) | |
| Observations | 29,757 | 29,757 |

Robust standard errors in parentheses. *** $p < 0.01$.

## 6. Concluding Remarks

In this study, we used the National Financial Capability Study (NFCS 2012–2018) data to examine the determinants of investment in financial securities. Of interest is how generational cohort and the perception of risk tolerance influence individual decision-making with regard to the ownership of financial assets. The NFCS dataset is best suited for this study, as it is nationally representative and has quite a large sample size of over 27,000 in each wave. Other than generational differences and risk tolerance, we investigate the effects of other factors, such as the level of educational attainment, the level of financial literacy, and household incomes, and how these variables correlate with the ownership of financial securities. The analytical method employed is a probit model with and without sample selection. Further, we account for the possible endogeneity of risk tolerance by estimating a bivariate probit model.

All of the predictor variables are estimated to have statistically significant effects on the ownership of financial securities, with the expected sign effects. There is clearly a generational cohort effect, whereby more Baby Boomers are predicted to own financial investments than Millennials, controlling for other factors such as incomes, education, and financial literacy, etc. Generation Xers are statistically less likely than Millennials to hold investments in financial assets. With regard to the effects of risk tolerance, we observed that individuals who have a higher tolerance for risk (risk-loving) tend to have a higher likelihood of investing in financial securities. Educational level, financial literacy, home ownership, and having an emergency fund are all correlated positively with having investments in financial assets. Finally, Baby Boomers (Gen Xers) are less (more) risk-tolerant than Millennials, accounting for levels of education and incomes.

The study has a few limitations, a couple of which are worth mentioning. While the study uses pooled cross-sectional data, which is the best available, a longitudinal dataset would allow for tracking of the same-aged cohorts over time to assess any real changes in their attitudes to financial investments as they age. The cohort effects that we report may be argued to be static effects that might change over time. Second, the NFCS survey is administered online, which introduces biases due to the self-selection of individuals of certain demographic characteristics and the exclusion of others. While we have controlled

for selection biases, it can be argued that not all individuals have the ability to participate in online surveys: there are still pockets within the population without access to the Internet, or who lack the time or skills to respond to online surveys. Such individuals might not be represented in the surveys, thus, limiting the generalizability of the findings to all segments of the population.

Further research in this area will be critical in addressing some of the limitations outlined above. In particular, the paper raises the conundrum of Baby Boomers being less risk-loving, but at the same time more likely to own investments in financial securities. This apparent contradiction will need to be studied further to uncover any confounding variables that may be influencing this result. The evidence reported in this paper is based on the United States; it would be interesting to see whether the relationship of generational attitudes and financial investment reported in this paper holds true in other countries around the world.

**Author Contributions:** Conceptualization, J.A.; Data Curation, J.A.; Formal Analysis, C.B.N.; Methodology, C.B.N.; Software, C.B.N.; Formal analysis. C.B.N.; Validation, C.B.N.; Visualization, J.A.; Writing—Original Draft, J.A.; Writing—Review and Editing, J.A. and C.B.N. All authors have read and agreed to the published version of the manuscript.

**Funding:** This research received no external funding.

**Institutional Review Board Statement:** Not applicable.

**Informed Consent Statement:** Not applicable.

**Data Availability Statement:** Dataset publicly available at Financial Capability Study (available on: usfinancialcapability.org; accessed on 25 April 2022).

**Acknowledgments:** The authors thank the editors and three anonymous reviewers for helpful suggestions and comments.

**Conflicts of Interest:** The authors have no financial or non-financial conflict of interest directly or indirectly related to this work.

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
