# Peer review of "Generational Differences, Risk Tolerance, and Ownership of Financial Securities: Evidence from the United States"

_ijfs, doi:10.3390/ijfs10020035_

Round 1
Reviewer 1 Report
Dear Authors
I thoroughly enjoyed reading your paper, and believe that it has potential to contribute to extant research. Thank you for your research.
Nevertheless, I feel you need to consider and address some aspects:
1- Please update significantly your literature references. Most cited doccuments are more than 5 years old and this reasearch area has been under significant evolution in the short past years. Also, make sure you also compare your results to those of more recent research, to stress the importance of your contribution;
2 - Please clarify why you feel that the research design is adequate in terms of grouping your independent variables (income; education;...; but most of all, "generation cohorts"). By grouping categorical variables further, you are significantly reducing the variance in data (and thus informativness). Can you share your econometric results using original measurement of those variables?
3 - A significant number of additional controls used in previous literature are missing from your models (e.g. gender). Can you review them and discuss the implication of that?
4 - Your measure of Financial Knowledge/ literacy is subjective and based on individuals' self perception. Literature shows that overconfident individuals (that evaluate themselves as more knowledgeable), are more risk tolerant. How do you consider this in your results?
5 - Finally, your classification of babyboomers (large older cohort) are significantly less tech savvy, which might asymetrically affect the online survey participation. Perhaps your results in terms of generational cohorts investment behaviors (with puzzling behavior from this group), is significantly justified by the former.
On some minor notes:
- review your equations: some are pictures, and ought to be typped; eq 12 are actually 2 equations;...
- place the tables next to the text, as is common in this journal;
- page 8 (lines 300 and on) is repeated information;
- some minor typo and extra space in text
Kind regards
Reviewer 2 Report
Referee report of ijfs-1667269
Summary
This paper discusses how generational attitudes and risk tolerance affect people’s choice to invest in financial securities. Through the probit model, it is confirmed that compared with Millennials, generation Xers are less likely to invest in financial securities, while baby boomers are more likely to invest. This topic is interesting and meaningful, but the research methods are not rigorous and some conclusions are confusing.
General comments
- Probit models alone are not enough to study generational differences in investment decisions. The authors need to supplement the ANOVA test to verify the generational differences.
- The conclusion that “baby boomers (gen Xers) are less (more) risk tolerant than millennials, accounting for levels of education and incomes” is confusing.
2.a. Firstly, this result is contradictory to another conclusion in this paper that individuals with higher risk tolerance are more likely to invest in financial securities. Authors attempt to explain this conundrum in terms of educational attainment and income, but lack sufficient empirical evidence.
2.b. Additionally, to analyze the effects of educational attainment, income, financial knowledge, and risk tolerance across generations, authors should include interaction terms in the regression or use a hierarchical regression.
2.c. Finally, it is better to find a reasonable explanation for this conclusion.
- The bivariate probit model requires that the partial observability model uses only the information from the product of the two dependent variables. In this paper, the first dependent variable () is represented by the outcome of financial securities ownership, the second dependent variable () is represented by the outcome of high risk tolerance, and the partial observability () is also represented by the outcome of financial securities ownership. It is hard to judge whether the partial observability in this paper meets the requirements of the bivariate probit model, that is, . Authors are advised to add descriptive statistics to illustrate.
- The observations of some variables are inconsistent in the two descriptive statistics (Table 1.a and Table 1.b). For example, the observation of risk tolerance with the original survey data is 77716, while the observation of risk tolerance with the variables constructed (Low/Medium/High risk tolerance) is 80164. It would be better to explain how the extra 2448 observations are constructed.

Reviewer 3 Report
The presented scientific study is interesting and important. However, it contains a number of shortcomings that need to be addressed.
1. In the abstract it is necessary to state the used scientific research methods, to clearly define the goal and results of the work.
2. In conclusion, it is also necessary to clearly state the results and clearly respond to the objectives set. It is also necessary to indicate the direction in which further research on this issue by the authors will take.
3. I consider the amount of literature used to be insufficient, I recommend supplementing the references with several current sources, for example:
Peráček, T. (2021). A few remarks on the (im) perfection of the term securities: a theoretical study. Juridical Tribune - Tribuna Juridica, 11 (2), pp. 135-149, doi: 10.24818 / TBJ / 2021/11 / 2.01
Korauš, A .; Dobrovič, J .; Polak, J .; Kelemen, P. 2019. Security position and detection of unusual business operations from science and research perspective, Entrepreneurship and Sustainability Issues 6 (3): 1270-1279. https://doi.org/10.9770/jesi.2019.6.3(15)
Mucha, B. (2021). Evaluation of the State of Implementation of the European Structural and Investment Funds: Case Study of the Slovak Republic. Online Journal Modeling the New Europe, 35, 4-24, doi: 10.24193 / OJMNE.2021.35.0
Jankelová, N., Jankurová, A., Beňová, M. & Skorková, Z. (2018) Security of the business organizations as a result of the economic crisis. Entrepreneurship and Sustainability Issues, 5 (3), pp. 659–671, doi: 10.9770 / jesi.2018.5.3 (18)
Klimek, L. & Funta, R. (2021). Data and E-commerce: An Economic Relationship. Danube, 12 (1), pp. 33–44, doi: 10.2478 / danb-2021-0003
Round 2
Reviewer 1 Report
Dear Authors
Thank you for addressing the comments provided. I believe you acknowledge and provide answers for most of it.
I would still reccomend that you would provide information in the paper about the additional robustness analysis you presented in the responde to my comments. If not the tables, at least mention the results.
Kind regards
Reviewer 2 Report
To my mind, the existing version is acceptable.
Author Response
Thank you!
Reviewer 3 Report
The authors made the required changes to a reasonable extent, but in the revised document they did not indicate some of the declared changes as an extension of the abstract or the focus of further research in the future.
They also did not accept the proposed supplementation of the literature with works by authors, which, although indirectly related to the researched issues.
Personally, I would again recommend in the article to use the knowledge from the works of the team of authors of Western Europe, such as:
Spaeth, W., Peracek, T. (2022). Cryptocurrencies, Electronic Securities, Security Token Offerings, Non Fungible Tokens: New Legal Regulations for “Crypto Securities” and Implications for Issuers and Investor and Consumer Protection. In: Kryvinska, N., Greguš, M. (eds) Developments in Information & Knowledge Management for Business Applications. Studies in Systems, Decision and Control, vol 420. Springer, Cham. https://doi.org/10.1007/978-3-030-95813-8_10
Rajnoha, R., Lesníková, P., Korauš, A. From financial measures to strategic performance measurement system and corporate sustainability: Empirical evidence from Slovakia. Economics and Sociology, 2016, 9 (4), pp. 134–152
Chovancová, B., Árendáš, P., Slobodník, P., Vozňáková, I. Country risk at investing in capital markets - The case of Italy. Problems and Perspectives in Management, 2019, 17 (2), pp. 440–448
